# Chronic Kidney Disease Induced by Cadmium and Diabetes: A Quantitative Case-Control Study

**DOI:** 10.3390/ijms24109050

**Published:** 2023-05-20

**Authors:** Supabhorn Yimthiang, David A. Vesey, Phisit Pouyfung, Tanaporn Khamphaya, Glenda C. Gobe, Soisungwan Satarug

**Affiliations:** 1Occupational Health and Safety, School of Public Health, Walailak University, Nakhon Si Thammarat 80160, Thailand; ksupapor@mail.wu.ac.th (S.Y.); phisit.po@mail.wu.ac.th (P.P.); tanaporn.kh@mail.wu.ac.th (T.K.); 2The Centre for Kidney Disease Research, Translational Research Institute, Brisbane 4102, Australia; david.vesey@health.qld.gov.au (D.A.V.); g.gobe@uq.edu.au (G.C.G.); 3Department of Kidney and Transplant Services, Princess Alexandra Hospital, Brisbane 4102, Australia; 4School of Biomedical Sciences, The University of Queensland, Brisbane 4072, Australia; 5NHMRC Centre of Research Excellence for CKD QLD, UQ Health Sciences, Royal Brisbane and Women’s Hospital, Brisbane 4029, Australia

**Keywords:** albuminuria, β_2_-microglobulin, cadmium, diabetes, diabetic nephropathy, GFR, tubular proteinuria

## Abstract

Kidney disease associated with chronic cadmium (Cd) exposure is primarily due to proximal tubule cell damage. This results in a sustained decline in glomerular filtration rate (GFR) and tubular proteinuria. Similarly, diabetic kidney disease (DKD) is marked by albuminuria and a declining GFR and both may eventually lead to kidney failure. The progression to kidney disease in diabetics exposed to Cd has rarely been reported. Herein, we assessed Cd exposure and the severity of tubular proteinuria and albuminuria in 88 diabetics and 88 controls, matched by age, gender and locality. The overall mean blood and Cd excretion normalized to creatinine clearance (C_cr_) as E_Cd_/C_cr_ were 0.59 µg/L and 0.0084 µg/L filtrate (0.96 µg/g creatinine), respectively. Tubular dysfunction, assessed by β_2_-microglobulin excretion rate normalized to C_cr_(E_β2M_/C_cr_) was associated with both diabetes and Cd exposure. Doubling of Cd body burden, hypertension and a reduced estimated GFR (eGFR) increased the risks for a severe tubular dysfunction by 1.3-fold, 2.6-fold, and 84-fold, respectively. Albuminuria did not show a significant association with E_Cd_/C_cr_, but hypertension and eGFR did. Hypertension and a reduced eGFR were associated with a 3-fold and 4-fold increases in risk of albuminuria. These findings suggest that even low levels of Cd exposure exacerbate progression of kidney disease in diabetics.

## 1. Introduction

Cadmium (Cd) is a toxic metal pollutant of global public health concern because even low environmental exposure to this metal promotes high prevalence of diseases, such as chronic kidney disease (CKD) and type 2 diabetes [1]. Prediabetes and diabetes are defined as fasting plasma glucose ≥ 110 mg/dL and 126 mg/dL, respectively. Evidence that environmental exposure to Cd raises fasting blood glucose is apparent from studies in many countries including the U.S. [2,3,4,5,6,7]. The Wuhan–Zhuhai prospective cohort study reported a particularly strong association between Cd exposure and pre-diabetes. Over 3-year period, it reported that for each tenfold increase in urinary Cd, the prevalence of prediabetes increased by 42% [5].

The global prevalence of diabetes, and its major complication, diabetic kidney disease (DKD), have reached epidemic proportions. Of note, DKD, marked by albuminuria and a rapid reduction in estimated glomerular filtration rate (eGFR) is the leading cause of kidney failure, (eGFR below 15 mL/min/1.73 m^2^), which requires dialysis or kidney transplantation for survival. In cross-sectional and prospective cohort studies of 231 diabetic patients in the Netherlands, both Cd and active smoking were associated with a progressive decline in eGFR [8,9]. These studies support the premise that exposure to even low levels of environmental Cd promote the development and progression of DKD.

Cd accumulates primarily in the proximal tubular epithelial cells of the kidneys, where the burden of Cd as µg/g kidney tissue weight increases with age [10,11]. Approximately 0.001–0.005% of Cd in the body is excreted in the urine each day, and the biological half-life of Cd in the kidney cortex is estimated to be 30 years for non-smokers [12]. Tubular proteinuria, indicated by an increase in the excretion of the low-molecular weight protein, β_2_-microglobulin (β_2_M) ≥ 300 µg/g creatinine has been used to indicate Cd-induced kidney disease for many decades [13]. It is noteworthy that exposure to environmental Cd has also been linked to a reduction eGFR in the general population [14,15,16], but reports of tubular proteinuria in people with diabetes who are exposed to Cd is limited. Of concern, data from the 1990 Belgian population study (Cadmibel) first suggested that people with diabetes were susceptible to Cd-induced nephrotoxicity [17]. Similar observations were then made in studies from Sweden [18,19], Australia [20], the U.S. [14], and Korea [21].

The present study aimed to evaluate the effects of diabetes and Cd exposure on kidney outcomes. Thus, we compared Cd exposure levels, tubular proteinuria, albuminuria, and the reduction of eGFR in diabetics and non-diabetics matched by age, gender and residential location. We used the excretion of Cd as a measure of long-term exposure, (body burden), and a sign of its nephrotoxicity. The utility of Cd excretion as an indicator of its toxicity to kidneys is based on a study showing that Cd excreted in complexes with metallothionein (MT) emanated from injured or dying kidney tubular epithelial cells [1,22]. We employed established equations of the Chronic Kidney Disease Epidemiology Collaboration (CKD-EPI) to compute the estimated GFR (eGFR) [23].

For the accurate quantification of the kidney burden of Cd and its effects, we normalized the excretion of Cd, albumin (Alb) and β_2_M (E_Cd_, E_Alb_, and E_β2M_) to creatinine clearance (C_cr_), denoted as E_Cd_/C_cr_, E_Alb_/C_cr_ and E_β2M_/C_cr_, respectively [19,20]. This C_cr_ normalization depicts an amount of a given chemical excreted per volume of filtrate, which is at least roughly related to the amount of the chemical excreted per nephron. In effect, C_cr_ normalization is unaffected by creatinine excretion, but it corrects for differences in the number of surviving nephrons among study subjects [24].

## 2. Results

### 2.1. Characteristics of the Study Subjects

This case-control study consisted of 88 non-diabetics, and 88 diabetics of which 48 and 37 had diabetes for less than 10 years or more than 10 years, respectively (Table 1).

Cases and controls were matched by age, gender, and residential locality. The mean age and percentages (%) of females in cases and controls did not differ. The % smoking, hypertension, obesity, and a reduced eGFR across three groups did not differ. The mean BMI, mean blood Cd, mean E_Cd_/C_cr_, mean E_Cd_/E_cr_, and mean eGFR were similar in the three groups. The mean duration of diabetes in the <10-yr and ≥10-yr groups were 4.2 and 15.8 years, respectively.

Mean E_β2M_/C_cr_, mean E_Alb_/C_cr_ were higher in cases than controls as was the % of microalbuminuria, defined as (E_Alb_/C_cr_) × 100 ≥ 20 mg/L filtrate. The % of microalbuminuria was lowest in controls, middle in the <10-yr DM and highest in ≥10-year DM groups. The % of microalbuminuria was increased from 10.7% in controls to 33.3% and 45.7% in the <10-yr DM and ≥10-year DM groups, respectively. Macroalbuminuria was not found in controls, but was present in 11.4–13.3% of those with DM. The overall % of mild and severe tubular dysfunction, defined as (E_β2M_/C_cr_) × 100 ≥ 300 and ≥1000 µg/L filtrate [25] were 66.2%, and 32.4%, respectively. More than half (57.7%) of the ≥10-DM group has severe tubular proteinuria.

Quantitative results obtained from E_cr_-normalized and C_cr_-normalized data were similar with respective to mean E_Cd_/E_cr_, mean E_β2M_/E_cr_ and men E_Alb_/E_cr_. However, the E_cr_-normalized data gave lower % of microalbuminuria and macroalbuminuria in controls and diabetics, compared to C_cr_-normalized data. The overall % of mild and severe tubular dysfunction, defined as E_β2M_/E_cr_ ≥ 300 and ≥1000 µg/g creatinine [26,27] were 70.5%, and 37.4%, respectively. More than half (65.5%) of the ≥10-DM group has severe tubular defect.

### 2.2. Independent Associations of Cadmium and Diabetes with Measurement of Kidney Function

We next undertook multivariate analyses to evaluate independent associations of Cd and diabetes with quantitative kidney tubular and glomerular functions reflected by the β_2_M excretion rate, Alb excretion rate and eGFR. A set of eight independent variables incorporated into the regression model included age, BMI, log_10_[Cd]_b_, log_2_[(E_Cd_/C_cr_) × 10^5^], smoking, gender, hypertension, and diabetes (Table 2).

Eight independent variables together accounted for 33%, 17.1% and 14.1% of the variation in the β_2_M excretion rate (*p* < 0.001), Alb excretion rate (*p* = 0.001) and reduction in eGFR (*p* = 0.001). The excretion rate of β_2_M as log[(E_β2M_/C_cr_) × 10^3^] was associated with age (β = 0.164), urine Cd (β = 0.244) and diabetes (β = 0.432). The other 5 independent variables did not show a significant association with the β_2_M excretion. The excretion rate of Alb as log [(E_Alb_/C_cr_) × 10^4^] was associated with hypertension (β = 0.259), and diabetes (β = 0.286). In contrast, eGFR was inversely associated with age (β = −0.357) only.

### 2.3. Effects of Cadmium and Diabetes on Risks of Adverse Kidney Outcomes

We used multivariable logistic regression to determine the prevalence odds ratios (POR) for three adverse kidney outcomes associated with Cd and duration of diabetes. The adverse outcomes of kidney function assessed were a defective tubular re-absorption of β_2_M, microalbuminuria, and a reduced eGFR, indicated by (E_β2M_/C_cr_) × 100 ≥ 300 µg/L, (E_Alb_/C_cr_) × 100 ≥ 20 mg/L, and eGFR ≤ 60 mL/min/1.73 m^2^, respectively (Table 3).

Age, log_2_[(E_Cd_/C_cr_) × 10^5^] and ≥10-y DM duration were associated with increased risk of tubular dysfunction. Per a 1-year increase in age and doubling urine Cd, the POR for tubular dysfunction rose by 6% (*p* = 0.029), and 26% (*p* = 0.044) respectively. Compared to controls, the POR for tubular dysfunction was increased by 4-fold in the ≥10-yr DM group (*p* = 0.036). The POR for microalbuminuria was associated only with DM duration, where there was a 6.142 (95% CI: 2.004, 18.83, *p* = 0.001) in those who had DM ≥10 years, compared to controls. The POR for a reduced eGFR was increased by 14.7% (*p* = 0.029) per a 1-year increase in age (*p* = 0.001). There was a 6.9-fold (*p* = 0.009) increase in POR for reduced eGFR in the ≥10 yr-DM group (*p* = 0.009).

### 2.4. Quantitation of Effects of Cadmium and Diabetes on Kidney Tubular Reabsorption of β_2_M

Doubling of Cd body burden and hypertension increased the POR for a severe tubular dysfunction to 1.317 and 2.621, respectively, while eGFR levels of 61−89 and ≤60 mL/min/1.73 m^2^ were associated with increases in the POR for a severe tubular proteinuria by 14-fold and 84-fold, compared with eGFR ≥ 90 mL/min/1.73 m^2^ (Table 4).

We further evaluated the effects of Cd and diabetes by scatterplots and covariance analyses, where the mean β_2_M excretion rates were derived for controls and diabetics adjusted for covariates. Results of these quantitative effect investigations are provided in Figure 1.

Log [(E_β2M_/C_cr_) × 10^3^] showed a moderate positive association with Cd body burden measured as log [(E_Cd_/C_cr_) × 10^5^] in controls (R^2^ = 0.151) and diabetics (R^2^ = 0.110) (Figure 1a). Data from a covariance analysis confirms a dose-effect relationship between β_2_M excretion rate and Cd body burden in both control and DM groups (Figure 1b). After adjustment for age, BMI and interactions, Cd body burden accounted for 12.2% and 17.3% of the variation β_2_M excretion rate in control and DM groups, respectively.

Equivalent covariance analysis of Alb excretion rate and eGFR indicate that the associations of these two parameters with Cd body burden were insignificant.

### 2.5. Quantitation of Effects of GFR on Microalbuminuria

An overall effect of GFR on microalbuminuria was assessed by a logistic regression analysis, where eGFR was categorized into three levels; ≥90, 61−89, ≤60 mL/min/1.73 m^2^. Among eight independent variables, only hypertension and eGFR levels were associated with the increment of POR for microalbuminuria.

The POR values for albuminuria increased by 3.3-, 4.5- and 4.6-fold in those with hypertension and groups with eGFR 61−89, and ≤60 mL/min/1.73 m^2^ (Table 5).

The relationships between eGFR excretion of β_2_M and Alb were assessed with scatterplots and covariance analyses. In these investigations, we compared mean β_2_M excretion and mean Alb excretion rates in controls and diabetics with adjustment for age, BMI and interaction. Results are provided in Figure 2.

Log[(E_β2M_/C_cr_) × 10^3^] showed a moderate inverse association with eGFR in controls (R^2^ = 0.216), while showing a strong inverse association with eGFR in diabetics (R^2^ = 0.461) (Figure 2a). A covariance analysis showed a dose-effect relationship between β_2_M excretion rate and eGFR in both control and DM groups (Figure 2b). After adjustment for age, BMI and interactions, eGFR accounted for 37.6% and 42.6% of the variation β_2_M excretion rate in control and DM groups, respectively.

Log [(E_Alb_/C_cr_) × 104] was inversely associated with eGFR only in the DM group (R^2^ = 0.197) in bivariate analysis (Figure 2c). However, after adjustment for age, BMI and interactions, the relationship between Alb excretion rate and eGFR in the DM group became insignificant (Figure 2d).

### 2.6. Quantification of Effects of Diabetes Duration and Hypertension on Alb Excretion Rate

To assess effects of duration of diabetes and hypertension, we compared mean Alb excretion rates in controls, <10-yr DM, and ≥10-y DM groups two diabetics. Results are provided in Figure 3.

The mean values of log[(E_Alb_/C_cr_) × 10^4^] were higher the <10-y and the ≥10-y DM groups, compared to controls (Figure 3a). After adjustment for covariates and interaction, mean log[(E_Alb_/C_cr_) × 10^4^] was higher in ≥10-y DM with hypertension than the hypertensive only control group.

## 3. Discussion

Environmental Cd exposure experienced by the diabetic and the non-diabetic groups could be considered as low to moderate, evident from the arithmetic means (SD) for blood Cd, E_Cd_/C_cr_ and E_Cd_/E_cr_ of 0.59 (0.74) µg/L, 0.0084 (0.0166) µg/L filtrate and 0.96 (1.83) µg/g creatinine respectively. However, these low environmental Cd exposures are associated an adverse outcome (tubular proteinuria) in both diabetic and non-diabetic control groups (Table 2 and Table 3). The risk of tubular proteinuria was increased by 26% (*p* = 0.044) per doubling of E_Cd_/C_cr_, an indicator of cumulative lifetime exposure or body burden. This pathologic outcome was increased by 4-fold in the diabetic group with an average duration of 15 years (*p* = 0.036, Table 3).

The percentage of a reduced eGFR in controls, <10-y DM and ≥10-y DM groups were 11.4%, 14.6% and 27.0%, respectively (Table 1). The prevalence of a reduced eGFR in the controls of 11.4% is higher than that reported in studies from Spain (7%) [28] and Taiwan (6.3%) [29], but in line with the global prevalence of CKD, which varies between 8% and 16% [30]. None of diabetic cases with reduced eGFR received special treatments other than antidiabetic medication (metformin). Due to a uniform treatment and uniform Cd exposure levels evident from the similar blood and urinary excretion rate across three groups, the higher number of cases with a reduced eGFR in the >10-y DM, compared with the < 10-y DM groups could be attributable to Cd and diabetes.

Evidence linking tubulopathy due to Cd accumulation and diabetes comes from covariance analyses (Figure 1), where a dose-effect relationship between E_β2M_/C_cr_ and E_Cd_/C_cr_ was seen in both controls and diabetics. Approximately 12% and 17% of the variation in E_β2M_/C_cr_ among non-diabetics and diabetics were attributed to Cd after adjustment for age, BMI and interactions. Thus, diabetes may have contributed to 5% of the of the E_β2M_/C_cr_ variation while another 12% was due to Cd. The mean E_β2M_/C_cr_ in the diabetics of the high E_Cd_/C_cr_ tertile was higher, compared to those in the lowest tertile (Figure 1b).

As scatterplots indicated, an inverse relationship between eGFR and E_β2M_/C_cr_ was particularly strong in the diabetics (R^2^ = 0.461) than controls (R^2^ = 0.216) (Figure 2a). An analysis with adjustment for covariates has revealed that 42.6% of E_β2M_/C_cr_ variation among the diabetics was attributable to the decrease of eGFR. This is a large proportion of the E_β2M_/C_cr_ variation that was explained by a single variable (eGFR). In the non-diabetic control group, eGFR explained 37.6% of the variation in β_2_M excretion rate. Thus, an inverse relationship between E_β2M_/C_cr_ and eGFR was universal.

A strong influence of eGFR levels on E_β2M_/C_cr_ was indicated also in a logistic regression analysis (Table 4), where a reduced eGFR (≤60 mL/min/1.73 m^2^) was associated with an 84-fold increase in risk for a severe tubular proteinuria, defined as (E_β2M_/C_cr_) × 100 ≥ 1000 µg/L filtrate. The risk of this tubular functional defect was increased by 14-fold in those with eGFR values ranging between 61−89 mL/min/1.73 m^2^, compared with eGFR ≥ 90 mL/min/1.73 m^2^.

The protein β_2_M has a molecular weight in low range (11,800 Da), and is thus filtered freely by the glomeruli, and is reabsorbed almost completely by the kidney’s tubular epithelial cells [13]. Excessive β_2_M excretion has most frequently been interpreted to be a result of defective tubular re-absorption of proteins from the ultrafiltrate [13,31,32,33]. It is noteworthy, however, that an enhanced β_2_M excretion can also be a consequence of nephron loss for any reasons [13,33]. When the reabsorption rate of β_2_M per nephron remains constant, its excretion will vary directly with its production. If the production and reabsorption per nephron remain constant as nephrons are lost, the excretion of β_2_M will rise [34]. Accordingly, a severe tubular proteinuria, which was 84-fold more prevalent in those with a reduced eGFR could be a consequence of nephron loss due to both Cd and diabetes.

Albumin is not normally filtered by glomeruli, due to its large molecular weight and its negative charge. The small amount of albumin which does enter the urinary space is reabsorbed almost completely and a fraction of this is returned to the systemic circulation [35,36]. In an experimental study, Cd was found to disable the cubilin/megalin receptor system of albumin endocytosis, leading to albuminuria [37]. In addition, Cd diminished expression of megalin and ClC5 channels [38]. Cd may also increase glomerular permeability to albumin, as shown in another study, where a non-cytotoxic concentration of Cd (1 µM) increased the permeability of human renal glomerular endothelial cells in monolayers and caused the redistribution of the adherens junction proteins, vascular endothelial-cadherin and β-catenin [39,40].

Like β_2_M, the risk of albuminuria was increased as eGFR fell. Compared with eGFR ≥ 90 mL/min/1.73 m^2^, the risk of albuminuria was increased by 4.5-fold and 4.6-fold in those with eGFR 61−89, and ≤60 mL/min/1.73 m^2^, respectively (Table 5). However, unlike the excretion rate of β_2_M, E_Alb_/C_cr_ did not show a significant inverse associated with eGFR (Figure 2d), thereby suggesting a specific effect of diabetes. This is consistent with the literature reports suggesting that diabetes may cause injury to podocytes and the glomerular endothelial cells [41,42].

Hypertension was another risk factor for albuminuria in addition to eGFR reduction (Table 5). This was strengthened a covariance analysis with control for age and BMI, which showed higher mean E_Alb_/C_cr_ in those who had diabetes ≥ 10 years plus hypertension, compared to those with diabetes only (Figure 2c). Of relevance, a study of the indigenous population of the Torres Strait (Australia), where diabetes was highly prevalent, observed a strong association of E_Cd_/E_cr_ and albuminuria after controlling for age, gender, BMI, smoking status, and hypertension [20]. In the same study, age- and BMI- adjusted E_Cd_/E_cr_ was higher in people with diabetes plus albuminuria (1.91 μg/g creatinine), compared to those with diabetes without albuminuria (0.74 μg/g creatinine).

It is noteworthy that Cd exposure quantified with blood Cd, an indicator of exposures within the last three months [43], did not show a significant association with tubular proteinuria, albuminuria or a reduced eGFR in the present study (Table 2 and Table 3). This may have been due to a narrow range of both age and blood Cd levels, often encountered when the sample size is modest (*n* = 176). Indeed, blood Cd in ranges with participants in our study have been found to be associated with increases in risk of adverse kidney outcomes in other studies using a large samples size. For example, blood Cd levels ≥ 0.61 μg/L were associated with 1.8- and 2.2-fold increases in risk of a reduced eGFR and albuminuria in participants of the U.S. National Health and Nutrition Examination Survey (NHANES 2007–2012, n = 12,577) [14]. Cd-induced eGFR reduction was more pronounced in those with diabetes, hypertension or both [14]. Blood Cd levels ≥ 0.53 μg/L were associated, respectively with 2.21- and 2.04-fold increases in the risk of a reduced eGFR and albuminuria among participants in the U.S. NHANES 2011–2012 (n = 1545) [15]. Of a total 262 chemicals tested, blood Cd was associated with a reduced GFR, albuminuria, and a reduced GFR plus albuminuria among U.S. adults enrolled in NHANES 1999–2016 (n = 46,748) [44].

Both blood Cd and E_Cd_/E_cr_ have been linked to increased risk of albuminuria among those enrolled in NHANES 1999–2006 (n = 5426), where a 63% increase in risk of albuminuria was associated with E_Cd_/E_cr_ > 1 µg/g creatinine plus blood Cd > 1 µg/L, and a 48% increase in risk of a reduced eGFR was associated with blood Cd levels > 1 µg/L [16]. The reductions in eGFR due to Cd nephropathy have previously been attributed to glomerular injury. However, current evidence suggests that sufficient tubular cell injury disables glomerular filtration and leads to nephron atrophy, glomerulosclerosis, and interstitial inflammation and fibrosis [45,46,47]. In a histopathological examination of kidney biopsies from healthy kidney transplant donors [48], the degree of tubular atrophy was positively associated with the level of Cd accumulation. Tubular atrophy was observed at relatively low Cd levels [48].

The results of the present study have implicated Cd exposure in the pathogenesis of tubular proteinuria in both non-diabetic and diabetic groups. This tubular proteinuria was indicated by excessive excretion of the low molecular weight filterable protein, β_2_M is inversely associated with eGFR. In a previous study [49], an inverse association between eGFR and E_β2M_ was evident in people with low GFR (β = −0.332), but not in those with GFR >90 mL/min/1.73 m^2^ (β = −0.008). These data suggested Cd-induced nephron loss and reduced tubular reabsorption of β_2_M.

Intriguingly, in a Japanese prospective cohort study, tubular proteinuria (E_β2M_/E_cr_ ≥ 300 μg/g creatinine) was associated a 79% increased risk of a large decline eGFR (10 mL/min/1.73 m^2^) over a 5-year period [50]. An increase of E_β2M_/E_cr_ from 84.5 to ≥145 μg/g creatinine predicted hypertension [51]. In the general Japanese population, E_β2M_/E_cr_ combined with macroalbuminuria was a predictor of a high-rate of eGFR deterioration [52]. Among diabetic patients, those with a DM duration ≥15 years plus hypertension had a high-to-very high risk of CKD progression [53]. Data from the Framingham Heart Study (n = 7065) have linked higher plasma β_2_M levels to an increased risk of prevalent and incident hypertension [54]. These data demonstrate that β_2_M could be of utility in early detection of serious outcomes. Given that the global prevalence of diabetes has reached epidemic proportions, research into β_2_M excretion as an early warning sign of diabetic complications is warranted.

## 4. Materials and Methods

### 4.1. Recruitment of Cases and Controls

Diabetic cases were recruited together with age- and gender-matched non-diabetic controls from the health promotion center of Pakpoon Municipality, Nakhon Si Thammarat Province, Thailand. It was undertaken during June 2020 to May 2021. The inclusion criteria were resident in the Pakpoon municipality, 40 years of age or older who were diagnosed with type 2 diabetes or were apparently healthy. The exclusion criteria were non-residents of Pakpoon municipality, pregnancy, breast-feeding, hospital record or physician’s diagnosis of an advanced chronic disease. All subjects were provided with details of study objectives, study procedures, benefits, and potential risks, and they all provided their written informed consents prior to participation. The sociodemographic data, education attainment, occupation, health status, family history of diabetes, and smoking status were obtained by structured interview questionnaires. Diabetes was defined as plasma glucose [Glc]_p_ levels ≥ 126 mg/dL. Hypertension was defined as systolic blood pressure ≥ 140 mmHg, or diastolic blood pressure ≥ 90 mmHg. After excluding subjects with missing data, 176 subjects (88 diabetics and 88 apparently healthy, non-diabetic controls) were included in this study.

### 4.2. Blood and Urine Sampling and Analysis

Participants were requested to fast overnight, and the collection of blood and urine samples was carried out at a local health center of Pakpoon Municipality in the morning of the following day. For glucose assay, blood samples were collected in tubes containing fluoride as an inhibitor of glycolysis. Blood samples for Cd analysis were collected in separate tubes containing ethylene diamine tetra-acetic acid (EDTA) as an anticoagulant. The blood and urine samples were kept on ice and transported within 1 h to the laboratory of Walailak University, where plasma samples were prepared. Aliquots of urine, whole blood and plasma samples were stored at −80 °C for later analysis. Fasting plasma glucose concentrations ([Glc]_p_) were measured to ascertain diabetes diagnosis and diabetes free stage of controls. The assay of plasma concentration of glucose was based on colorimetry. Assays of creatinine in urine and plasma ([cr]_u_, [cr]_p_]) were based on the Jaffe reaction. Urine concentration of albumin ([Alb]_u_) was determined using an immunoturbidimetric method. The Beta-2 microglobulin matched ELISA antibody pair set, human (Sino Biological Inc., Wayne, PA, USA) was employed to determine urine concentration of β_2_M ([β_2_M]_u_) with the low detection limit of 3.13 pg/mL.

### 4.3. Quantiation of Cd in Blood and Urine Samples

Blood Cd concentration ([Cd]_b_, [Pb]_b_) was determined with the GBC System 5000 Graphite Furnace Atomic Absorption Spectrophotometer (GBC Scientific Equipment, Hampshire, IL, USA). Multielement standards were used to calibrate metal analysis (Merck KGaA, Darmstadt, Germany). Reference urine and whole blood metal control levels 1, 2, and 3 (Lyphocheck, Bio-Rad, Hercules, CA, USA) were used for quality control, analytical accuracy, and precision assurance. The analytical accuracy of metal detection was checked by an external quality assessment every 3 years. All test tubes, bottles, and pipettes used in metal analysis were acid-washed and rinsed thoroughly with deionized water. When a [Cd]_b_ level was less than its detection limits, the concentration assign was the detection limit divided by the square root of 2 [55]. Sixty-one subjects (34.6%) had [Cd]_b_ below the detection limit of 0.1 µg/L.

### 4.4. Normalization of E_Cd_, E_β2M_ and E_Alb_ to E_cr_ and C_cr_

E_x_ was normalized to E_cr_ as [x]_u_/[cr]_u_, where x = Cd; β_2_M or Alb; [x]_u_ = urine concentration of x (mass/volume); and [cr]_u_ = urine creatinine concentration (mg/dL). The ratio [x]_u_/[cr]_u_ was expressed in μg/g of creatinine.

E_x_ was normalized to C_cr_ as E_x_/C_cr_ = [x]_u_[cr]_p_/[cr]_u_, where x = Cd; β_2_M or Alb; [x]_u_ = urine concentration of x (mass/volume); [cr]_p_ = plasma creatinine concentration (mg/dL); and [cr]_u_ = urine creatinine concentration (mg/dL). E_x_/C_cr_ was expressed as the excretion of x per volume of filtrate [24].

### 4.5. Estimated Glomerular Filtration Rates (eGFRs)

The GFR is the product of nephron number and mean single nephron GFR, and in theory, the GFR is indicative of nephron function [56,57,58]. In practice, the GFR is estimated from established chronic kidney disease–epidemiology collaboration (CKD-EPI) equations and is reported as eGFR [23,59].

Male eGFR = 141 × [plasma creatinine/0.9]^Y^ × 0.993^age^, where Y = −0.411 if [cr]_p_ ≤ 0.9 mg/dL, and Y = −1.209 if [cr]_p_ > 0.9 mg/dL. Female eGFR = 144 × [plasma creatinine/0.7]^Y^ × 0.993^age^, where Y = −0.329 if [cr]_p_ ≤ 0.7 mg/dL, and Y = −1.209 if [cr]_p_ > 0.7 mg/dL. For dichotomous comparisons, CKD was defined as eGFR ≤ 60 mL/min/1.73 m^2^. CKD stages 1, 2, 3a, 3b, 4, and 5 corresponded to eGFR of 90–119, 60–89, 45–59, 30–44, 15–29, and <15 mL/min/1.73 m^2^, respectively.

### 4.6. Statistical Analysis

Data were analyzed with IBM SPSS Statistics 21 (IBM Inc., New York, NY, USA). The Kruskal–Wallis test was used to assess differences in means among three groups, and the Pearson chi-squared test was used to assess differences in percentages. The one-sample Kolmogorov–Smirnov test was used to identify departures of continuous variables from a normal distribution, and logarithmic transformation was applied to variables that showed rightward skewing before they were subjected to parametric statistical analysis. The multivariable logistic regression analysis was used to determine the Prevalence Odds Ratio (POR) for categorical outcomes. Obesity was designated when BMI > 30 kg/m^2^. Reduced eGFR was assigned when eGFR ≤ 60 mL/min/1.73 m^2^.

For C_cr_-normalized data, kidney tubular dysfunction was defined as (E_β2M_/C_cr_) × 100 ≥ 300 µg/L of filtrate [25]. Microalbuminuria and macroalbuminuria were defined as (E_Alb_/C_cr_) × 100 ≥ 20 and 200 mg/L of filtrate, respectively. For E_cr_-normalized data, tubular proteinuria was defined as E_β2M_/E_cr_ ≥ 300 µg/g creatinine [26,27]. Microalbuminuria was defined as E_Alb_/E_cr_ ≥ 20 and ≥30 mg/g creatinine in men and women, respectively. Macroalbuminuria was assigned, when E_Alb_/E_cr_ > 300 mg/g creatinine. The mean β_2_M excretion rate and mean Alb excretion rate adjusted for age, BMI, and interaction in groups of subjects were obtained by univariate/covariance analysis with Bonferroni correction in multiple comparisons. For all tests, *p*-values ≤ 0.05 were considered to indicate statistical significance.

## 5. Conclusions

The present study shows that environmental exposure to Cd is one of the contributing factors to tubular proteinuria in people with and without diabetes. Chronic exposure to even low levels of environmental Cd contributes, at least in part to diabetic macroalbuminuria. The tubular proteinuria in people with diabetes who are also exposed to Cd is a manifestation of defective kidney tubular protein reabsorption and nephron loss.

## Figures and Tables

**Figure 1 ijms-24-09050-f001:**
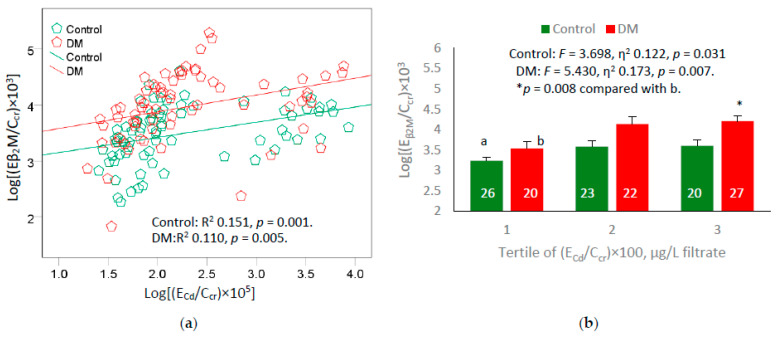
Comparing effects of body burden of cadmium and diabetes on β2M excretion rate. Scatterplot (**a**) relates log[(E_β2M_/C_cr_) × 10^3^] to log[(E_Cd_/C_cr_) × 10^5^] in non-diabetics and diabetics. Bar graph (**b**) depicts mean log[(E_β2M_/C_cr_) × 10^3^] for the non-diabetics and diabetics grouped by ranges of log[(E_Cd_/C_cr_) × 10^5^]. Coefficients of determination (R^2^) and *p*-values are provided for all scatterplots. Mean β_2_M excretion values were adjusted for age, BMI, and interactions. Mean (SD) values for (E_Cd_/C_cr_) × 100 tertiles 1, 2 and 3 are 0.046 (0.014), 0.109 (0.030), and 2.316 (2.197), µg/L of filtrate, respectively. The letters a and b denote the non-diabetic controls and the diabetics in the lowest tertile of (E_Cd_/C_cr_) × 100, respectively. The numbers of subjects are provided for all subgroups.

**Figure 2 ijms-24-09050-f002:**
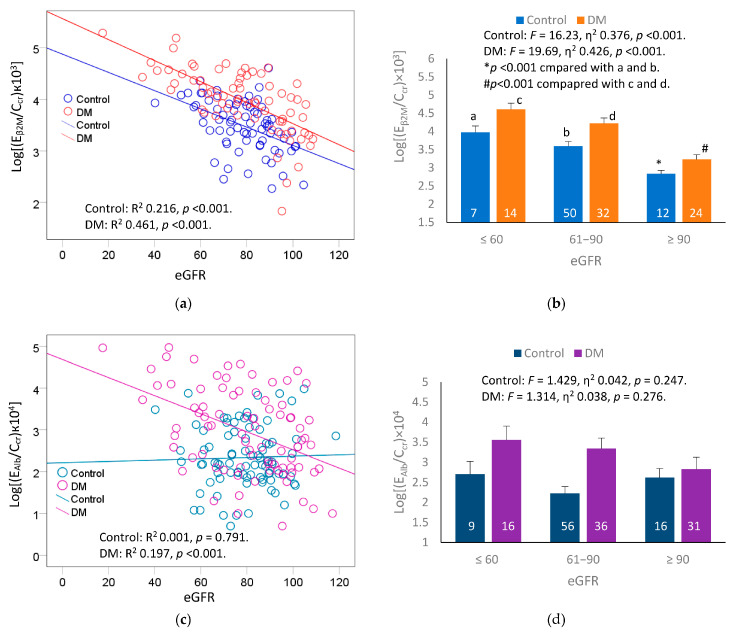
Comparing effects of eGFR on excretion rates of β_2_M and Alb. Scatterplot (**a**) relates log[(E_β2M_/C_cr_) × 10^3^] to eGFR in non-diabetics and diabetics. Bar graph (**b**) depicts mean log[(E_β2M_/C_cr_) × 10^4^] for the non-diabetics and diabetics grouped by eGFR ranges. Scatterplot (**c**) relates log[(E_Alb_/C_cr_) × 10^4^] to eGFR in non-diabetics and diabetics. Bar graph (**d**) depicts mean log[(E_Alb_/C_cr_) × 10^4^] for the non-diabetics and diabetics grouped by eGFR ranges. Coefficients of determination (R^2^) and *p*-values are provided for all scatterplots. Mean β_2_M excretion values were adjusted for age, BMI, and interactions. The letters a and b denote the non-diabetic controls with eGFR ≤ 60 and 61–90 mL/min/1.73 m^2^, re-spectively. The letters c and d denote the diabetics with eGFR ≤ 60 and 61–90 mL/min/1.73 m^2^, re-spectively. The numbers of subjects are provided for all subgroups.

**Figure 3 ijms-24-09050-f003:**
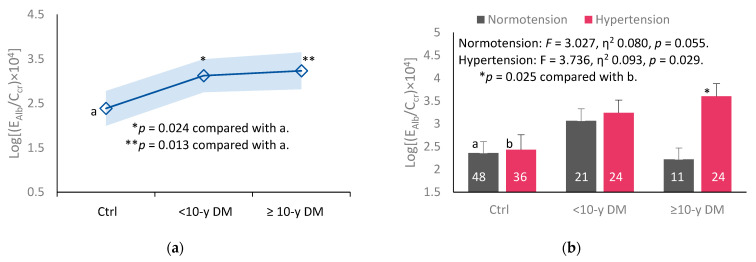
Comparing effects of diabetes and hypertension on albumin excretion rate. Graph (**a**) depicts mean log[(E_Alb_/C_cr_) × 10_4_] and the variation of each mean in control, <10-y DM and ≥ 10-y DM groups. Bar graph (**b**) depicts mean log[(E_Alb_/C_cr_) × 10^4^] for controls, <10-y DM and ≥10-y DM with and without hypertension. Coefficients of determination (R^2^) and *p*-values are provided for all scatterplots. Mean Alb excretion values were adjusted for age, BMI, and interactions. The letters a and b denote the controls with normotension and hypertension, respectively. The numbers of subjects are provided for all subgroups.

**Table 1 ijms-24-09050-t001:** Descriptive Characteristics of Non-diabetic Controls and Diabetic Cases.

Parameters	All Subjects *n* = 176	Non-DM *n* = 88	<10-yr DM *n* = 48	≥10-yr DM *n* = 37	*p*
Duration of diabetes ^a^, yrs	n/a	0	4.2 ± 2.1	15.8 ± 7.2	−
Fasting plasma glucose, mg/dL	131.7 ± 61.3	94 ± 12	177 ± 76	161 ± 55	<0.001
Blood Cd, µg/L	0.59 ± 0.74	0.64 ± 0.85	0.52 ± 0.61	0.57 ± 0.61	0.677
Female, %	80.9	80.7	79.2	83.9	0.863
Smoking, %	9.8	11.4	10.4	5.4	0.586
Hypertension, %	51.8	44.7	54.2	64.9	0.114
Age, years	59.9 ± 9.7	60.4 ± 9.2	59.2 ± 9.6	59.0 ± 11.2	0.489
BMI, kg/m^2^	25.4 ± 4.7	24.7 ± 4.4	26.3 ± 5.2	25.7 ± 4.6	0.098
Obese ^b^ (%)	10.4	5.7	18.8	10.3	0.058
eGFR ^c^, mL/min/1.73 m^2^	79.4 ± 18.0	79.4 ± 14.4	82.0 ± 19.6	77.6 ± 23.3	0.399
Reduced eGFR ^d^ (%)	15.6	11.4	14.6	27.0	0.086
Normalized to C_cr_ as E_x_/C_cr_ ^e^					
(E_Cd_/C_cr_) × 100, µg/L filtrate	0.84 ± 1.66	0.86 ± 1.69	0.53 ± 1.13	1.33 ± 2.21	0.248
(E_β2M_/C_cr_) × 100, µg/L filtrate	1313 ± 2396	543 ± 625	1975 ± 2914	2294 ± 3725	<0.001
(E_Alb_/C_cr_) × 100, mg/L filtrate	43.2 ± 127.7	8.8 ± 17.9	54.3 ± 99.2	112.4 ± 240.9	0.001
(E_β2M_/C_cr_) × 100 ≥ 300 µg/L (%)	66.2	54.2	78.0	80.8	0.008
(E_β2M_/C_cr_) × 100 ≥ 1000 µg/L (%)	32.4	15.3	46.3	57.7	<0.001
E_Alb_/C_cr_) × 100 ≥ 20 mg/L (%)	24.4	10.7	33.3	45.7	<0.001
E_Alb_/C_cr_) × 100, mg/L (%)					
<20	75.6	89.3	66.7	54.3	<0.001
20–199	18.3	10.7	20.0	34.3	<0.001
≥200	6.1	0	13.3	11.4	0.008
Normalized to E_cr_ as E_x_/E_cr_ ^f^					
E_Cd_/E_cr_, µg/g creatinine	0.96 ± 1.83	0.99 ± 1.94	0.69 ± 1.49	1.36 ± 2.08	0.376
E_β2M_/E_cr_, µg/g creatinine	1284 ± 1747	633 ± 762	2012 ± 2549	1856 ± 1589	<0.001
E_Alb_/E_cr,_ mg/g creatinine	41 ± 104	11 ± 24	60 ± 112	90 ± 173	<0.001
E_β2M_/E_cr_ ≥ 300 µg/g creatinine (%)	70.5	58.3	80.5	88.5	0.004
E_β2M_/E_cr_ ≥ 1000 µg/g creatinine (%)	37.4	20.8	48.8	65.4	<0.001
E_Alb_/E_cr_ ≥ 20 or ≥ 30 mg/g creatinine	20.7	7.1	33.3	37.1	<0.001
E_Alb_/E_cr_, mg/g creatinine (%)					
<30	79.3	92.9	66.7	62.9	<0.001
30–299	16.5	7.1	26.7	25.7	<0.001
≥300	4.3	0	6.7	11.4	0.001

*n*, number of subjects; DM, diabetics; BMI, body mass index; eGFR, estimated glomerular filtration rate; n/a, not applicable; E_x_, excretion of x; cr, creatinine; C_cr_, creatinine clearance; β_2_M, β_2_-microglobulin; Alb, albumin, Cd, cadmium; ^a^, among 88 diabetic cases, 85 had data on duration of diabetes; ^b^, obesity was assigned when BMI >30 kg/m^2^; ^c^ eGFR, was evaluated using the established CKD-EPI equations [23]. ^d^ Reduced eGFR was defined as eGFR ≤ 60 mL/min/1.73 m^2^. ^e^ E_x_/E_cr_ = [x]_u_/[cr]_u_; ^f^ E_x_/C_cr_ = [x]_u_[cr]_p_/[cr]_u_, where x = Cd, β_2_M or Alb [24]. Data for all continuous variables are arithmetic means ± standard deviation (SD). Data for urine Cd and urine Alb are from 141–167 subjects; data for all other variables are from 176 subjects. For each test, *p* ≤ 0.05 identifies statistical significance, determined by Pearson’s Chi-square test for percentage differences and the Kruskal–Wallis test for mean differences across three groups.

**Table 2 ijms-24-09050-t002:** Multivariate analyses of E_β2MG_/C_cr_, E_Alb_/C_cr_ and eGFR.

Independent Variables/Factors	Log[(E_β2M_/C_cr_) × 10^3^]			Log[(E_Alb_/C_cr_) × 10^4^]			eGFR		
	β	η^2^	*p*	β	η^2^	*p*	β	η^2^	*p*
Age	0.164	0.034	0.043	0.104	0.008	0.325	−0.357	0.131	<0.001
BMI	−0.141	0.028	0.065	−0.022	0.005	0.450	0.079	0.009	0.308
Log_10_[Cd]b	0.138	0.022	0.103	0.050	0.003	0.527	−0.077	0.006	0.383
Log_2_[(E_Cd_/C_cr_) × 10^5^]	0.244	0.077	0.002	0.033	0.000032	0.950	−0.104	0.010	0.272
Smoking	0.015	0.008	0.339	−0.036	0.002	0.622	0.012	0.000071	0.926
Gender (female)	−0.114	0.009	0.288	0.041	0.000025	0.955	0.089	0.010	0.269
Hypertension	0.048	0.024	0.088	0.259	0.036	0.035	0.023	0.001	0.732
Diabetes	0.432	0.083	0.001	0.286	0.042	0.023	−0.069	0.018	0.138
Interactions ^a^	−	0.038	0.031	−	0.053	0.010	−	0.038	0.031
Adjusted R^2^	0.330	−	<0.001	0.171	−	0.001	0.141	−	0.001

β, standardized regression coefficient; η^2^, eta squared; adjusted R^2^, coefficient of determination; β indicates strength of association of dependent variables with independent variables (first column). Adjusted R^2^ indicates the fraction of total variation of each dependent variable explained by all independent variables. η^2^ indicates the fraction of the variability of each dependent variable explained by a corresponding independent variable. ^a^, DM showed significant interactions with hypertension, gender, and smoking in regression analyses of E_β2M_/C_cr_, E_Alb_/C_cr_ and eGFR, respectively. For each test, *p*-values ≤ 0.05 indicate a statistically significant contribution of variation of an independent variable to variation of a dependent variable.

**Table 3 ijms-24-09050-t003:** Prevalence odds ratios for adverse kidney outcomes associated with cadmium and duration of diabetes.

Independent Variables/Factors	(E_β2M_/C_cr_) × 100 ≥ 300 µg/L		(E_alb_/C_cr_) × 100 ≥ 20 mg/L		eGFR ≤ 60 mL/min/1.73 m^2^	
	POR (95% CI)	*p*	POR (95% CI)	*p*	POR (95% CI)	*p*
Age	1.060 (1.006, 1.117)	0.029	1.038 (0.986, 1.094)	0.154	1.147 (1.058, 1.244)	0.001
BMI	0.945 (0.861, 1.038)	0.240	0.971 (0.888, 1.063)	0.527	0.981 (0.866, 1.110)	0.755
Log_10_[Cd]_b_	1.859 (0.976, 3.543)	0.059	1.235 (0.630, 2.422)	0.538	1.137 (0.475, 2.725)	0.773
Log_2_[(E_Cd_/C_cr_) × 10^5^]	1.260 (1.007, 1.577)	0.044	1.025 (0.843, 1.245)	0.807	1.138 (0.894, 1.449)	0.293
Smoking	2.643 (0.465, 15.04)	0.273	1.877 (0.124, 28.49)	0.650	1.877 (0.124, 28.49)	0.650
Gender (female)	2.043 (0.605, 6.900)	0.250	0.301 (0.085, 1.065)	0.062	0.448 (0.040, 4.989)	0.514
Hypertension	1.249 (0.521, 2.991)	0.618	2.317 (0.915, 5.865)	0.076	0.559 (0.165, 1.895)	0.350
Non-diabetics	Referent		Referent		Referent	
<10-yr DM	0.842 (0.209, 3.394)	0.809	2.315 (0.764, 7.016)	0.138	3.384 (0.736, 15.55)	0.117
≥10-yr DM	4.035 (1.094, 14.88)	0.036	6.142 (2.004, 18.83)	0.001	6.949 (1.613, 29.93)	0.009

eGFR, estimated glomerular filtration rate; POR, prevalence odds ratio; CI, confidence interval. (Eβ2M/Ccr) × 100 ≥ 300 µg/L, (EAlb/Ccr) × 100 ≥ 20 mg/L, and eGFR ≤ 60 mL/min/1.73 m^2^ are indicative of tubular dysfunction, microalbuminuria, and a reduced eGFR, respectively. Data were generated from logistic regression analyses relating the POR for the three indicators of adverse kidney outcomes to the seven independent variables listed in the first column. *p*-values < 0.05 indicate a statistically significant increase in the POR for adverse effects.

**Table 4 ijms-24-09050-t004:** Effects of cadmium and GFR on prevalence odds of a severe tubular dysfunction.

Severe Tubular Dysfunction ^a^					
Independent Variables/ Factors	β Coefficient (SE)	POR	95% CI		*p*
			Lower	Upper	
Age	0.010 (0.027)	1.010	0.957	1.066	0.705
BMI	0.000019 (0.055)	1.000	0.899	1.113	1.000
Log_10_[Cd]_b_	−0.317 (0.354)	0.728	0.364	1.457	0.370
Log_2_[(E_Cd_/C_cr_) × 10^5^]	0.276 (0.103)	1.317	1.077	1.612	0.007
Gender (female)	0.026 (0.742)	1.026	0.239	4.397	0.972
Hypertension	0.964 (0.476)	2.621	1.030	6.668	0.043
Smoking	0.598 (1.011)	1.819	0.251	13.185	0.554
eGFR, mL/min/1.73 m^2^					
≥90	Referent				
61−89	2.646 (0.730)	14.094	3.369	58.971	<0.001
≤60	4.430 (1.004)	83.958	11.742	600.342	<0.001

^a^, Severe tubular dysfunction is defined as (E_β2M_/C_cr_) × 100 ≥ 1000 µg/L filtrate; SE, standard error of mean; POR, prevalence odds ratio; CI, confidence interval. Data were generated from multivariable logistic regression analyses relating the POR for tubular dysfunction to eight independent variables (first column). *p*-values < 0.05 indicate a statistically significant increase in the POR for a defective tubular function.

**Table 5 ijms-24-09050-t005:** Effects of hypertension and GFR on prevalence odds of microalbuminuria.

Microalbuminuria ^a^					
Independent Variables/ Factors	β Coefficient (SE)	POR	95% CI		*p*
			Lower	Upper	
Age	0.002 (0.027)	1.002	0.951	1.056	0.950
BMI	0.039 (0.046)	1.039	0.950	1.138	0.401
Log_10_[Cd]_b_	0.144 (0.341)	1.155	0.592	2.251	0.672
Log_2_[(E_Cd_/C_cr_) × 10^5^]	0.024 (0.098)	1.025	0.845	1.242	0.804
Gender (female)	1.022 (0.617)	2.780	0.830	9.314	0.097
Hypertension	1.199 (0.463)	3.318	1.340	8.218	0.010
Smoking	0.885 (0.914)	2.423	0.404	14.531	0.333
eGFR, mL/min/1.73 m^2^					
≥90	Referent				
61−89	1.500 (0.596)	4.483	1.394	14.421	0.012
≤60	1.518 (0.725)	4.565	1.103	18.888	0.036

^a^, Microalbuminuria is defined as (E_Alb_/C_cr_) × 100 ≥ 20 mg/L filtrate; SE, standard error of mean; POR, prevalence odds ratio; CI, confidence interval; eGFR, estimated glomerular filtration rate. Data were generated from logistic regression analyses relating the POR for microalbuminuria to eight independent variables (first column). *p*-values < 0.05 indicate a statistically significant increase in the POR for microalbuminuria.

## Data Availability

All data are contained within this article.

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
