# Peer review of "Chronic Kidney Disease Induced by Cadmium and Diabetes: A Quantitative Case-Control Study"

_ijms, 2023, doi:10.3390/ijms24109050_

Round 1

Reviewer 1 Report

1) Exposure to cadmium (Cd) has been linked to an increased risk of diabetes (DM), but the evidence is limited and typically associated with higher levels of Cd in blood or urine.

2) The current study found that Cd levels of the subjects were low.

3) The aim of the research study was unclear. It was unclear whether the authors aimed to determine if low Cd levels have an additive effect on chronic kidney disease in DM or if chronic DM (>10 years) is a risk factor for Cd-associated chronic kidney disease.

4) This study may provide additional evidence that low cadmium levels contribute to kidney damage in long-term DM. However, it is important to note that the study did not control for variables such as HbA1c or diabetic drugs used. The authors used excretion of Cd as a measure of long-term exposure, but urinary cadmium levels can be influenced by recent exposure,  smoking status, and age. It is unclear how the authors controlled for these confounding factors.

5) Other biomarkers of cadmium exposure, such as cadmium concentrations in hair or nails, could be used to assess long-term exposure to cadmium.

Minor:

1) Line109: beta2MG should be consistent with the remaining.

2)Line 128: perhaps the authors meant to say eGFR was INVERSELY associated with age.

3) Lines 208 and 248; should be 0.216.

Author Response

Responses to Reviewer 1

We thank the Reviewer for knowledgeable comments and guidance to improve our manuscript.  We provide below point-by-point response to each comment.  Changes made to the text and new references are in blue.

Comments and Suggestions

1). Exposure to cadmium (Cd) has been linked to an increased risk of diabetes (DM), but the evidence is limited and typically associated with higher levels of Cd in blood or urine.

RESPONSE: We acknowledge that an effect of cadmium exposure on risk of diabetes has not been adequately studied.  However, in the present study, we focused on the effects of cadmium exposure and diabetes on kidney outcomes using a case-control design. The global prevalence of diabetes, and its major complication, diabetic kidney disease (DKD), have reached epidemic proportions. Epidemiologic studies implicating low environmental exposure to Cd on adverse kidney outcomes are abundant, but studies on these Cd-induced kidney pathologies in the diabetic population are limited.

2). The current study found that Cd levels of the subjects were low.

RESPONSE: We have inserted in the introduction previous studies that showed an adverse effect of Cd exposure on kidneys in people with diabetes as quoted below.

“Of concern, an increased susceptibility to Cd induced nephrotoxicity in people with diabetes was first noted in the 1990 Belgian population study (Cadmibel) [17]. Similar observations were then made in studies from Sweden [18,19], Australia [20], the U.S. [14], and Korea [22].”

[17] Buchet, J.P.; Lauwerys, R.; Roels, H.; Bernard, A.; Bruaux, P.; Claeys, F.; Ducoffre, G.; de Plaen, P.; Staessen. J.; Amery. A.; et al. Renal effects of cadmium body burden of the general population. Lancet 1990, 336, 699–702.

[18] Akesson, A.; Lundh, T.; Vahter, M.; Bjellerup, P.; Lidfeldt, J.; Nerbrand, C.; Samsioe, G.; Strömberg, U.; Skerfving, S. Tubular and glomerular kidney effects in Swedish women with low environmental cadmium exposure. Environ. Health Perspect. 2005, 113, 1627–1631.

[19] Barregard, L.; Bergstrom, G.; Fagerberg, B. Cadmium, type 2 diabetes, and kidney damage in a cohort of middle-aged women. Environ. Res. 2014, 135, 311–316.

[20] Haswell-Elkins, M.; Satarug, S.; O’Rourke, P.; Moore, M.; Ng, J.; McGrath, V.; Walmby, M. Striking association between urinary cadmium level and albuminuria among Torres Strait Islander people with diabetes. Environ. Res. 2008, 106, 379–383.

[21] Hwangbo, Y.; Weaver, V.M.; Tellez-Plaza, M.; Guallar, E.; Lee, B.K.; Navas-Acien, A. Blood cadmium and estimated glo-merular filtration rate in Korean adults. Environ. Health Perspect. 2011, 119, 1800–1805.

3). The aim of the research study was unclear. It was unclear whether the authors aimed to determine if low Cd levels have an additive effect on chronic kidney disease in DM or if chronic DM (>10 years) is a risk factor for Cd-associated chronic kidney disease.

RESPONSE: The study objective has been rewritten, as quoted below, to better reflect its content.

“The present study aimed to evaluate the effects of Cd exposure and diabetes on kidney outcomes. Thus, we compared Cd exposure levels, tubular proteinuria, albuminuria, and the reduction of eGFR in diabetics and non-diabetics matched by age, gender, and residential location.”

4). This study may provide additional evidence that low cadmium levels contribute to kidney damage in long-term DM. However, it is important to note that the study did not control for variables such as HbA1c or diabetic drugs used. The authors used excretion of Cd as a measure of long-term exposure, but urinary cadmium levels can be influenced by recent exposure, smoking status, and age. It is unclear how the authors controlled for these confounding factors.

RESPONSE: Although we have not obtained data on HbA1c, our assessment of adverse kidney outcomes was based on measured levels of fasting plasma glucose, and duration of DM. In theory, the length of DM is proportional to disease severity.  Diabetic cases were uniformly treated with antidiabetic medication. This is a common practice in Thailand based on a nationwide cohort study showing that glycemic control is the most effective treatment to protect against progression of CKD among diabetes patients [31]. Thus, it is likely that kidney outcome observed were due to Cd and/or diabetes.

[31] Sonthon, P.; Promthet, S.; Changsirikulchai, S.; Rangsin, R.; Thinkhamrop, B.; Rattanamongkolgul, S.; Hurst, C.P. The impact of the quality of care and other factors on progression of chronic kidney disease in Thai patients with type 2 diabetes mellitus: A nationwide cohort study. PLoS ONE 2017, 12, e0180977.

The POR values for adverse kidney outcomes (Tables 3, 4 and 5) all were adjusted for potential confounders that included age, BMI, smoking, hypertension, gender.  In quantitative analysis of adverse kidney outcomes in diabetics versus non-diabetic controls, the mean B2M excretion, mean Alb excretion were all adjusted for covariates (Figures 1, 2 and 3).

5). Other biomarkers of cadmium exposure, such as cadmium concentrations in hair or nails, could be used to assess long-term exposure to cadmium.

RSPONSE: The measurements of Cd contents in scalp hair and toenails as indicators of long-term Cd exposure are of utility in population-based studies involving large numbers of subjects.  In the present study, urinary and blood Cd levels are sufficient to identify exposure levels that are linked to kidney damage in non-diabetics and diabetics.   In this study where only a modest population was investigated, a comprehensive analysis was made which revealed clinically significant findings.

Minor:

1) Line109: beta2MG should be consistent with the remaining.

2)Line 128: perhaps the authors meant to say eGFR was INVERSELY associated with age.

3) Lines 208 and 248; should be 0.216.

RESPONSE: All the errors have been corrected.

Reviewer 2 Report

This case control study focused on diabetes patients and evaluated effects of cadmium and diabetes on kidney outcomes. Although with a modest population, the authors performed comprehensive analysis and concluded novel findings that are clinically significant. In addition to the advantages, there are few questions need to answer:

1.       The sample size of diabetic patients is limited, especially consider the number of patients with reduced eGFR. Are the patients with reduced eGFR undertaking specific treatments that could affect the results?

2.       What type of therapy treatment is given to the diabetes patients in the study? What could be the result if there are differences during treatment? If patients undertake dialysis, will there be any changes regarding the association between Cd and kidney function?

3.       Please provide reference indicating the cutoff number of the level of (Eβ2M/Ccr) ×100 regarding tubular dysfunction and severe tubular dysfunction in Table 3 and Table 4. The number of patients defined with severe tubular dysfunction is not summarized in the characteristic table.

4.       Table 3 indicates the POR of microalbuminuria is associated with DM duration whereas in table 5 both hypertension and eGFR levels are shown to be associated with microalbuminuria. How to interpret the differences?

5.       Figure 2 indicates Log [(Eβ2M/Ccr) ×103] is inversely associated with eGFR in both control and diabetics patients whereas Log [(EAlb/Ccr) ×104] dose not. What could be the reason of the differences? Does it suggest that Eβ2M/Ccr is a better indicator of eGFR level?

There are some minor spelling error ("Alb excretion rats", page 12)

and repeated content ("These findings" in abstract)

Author Response

Responses to reviewer 2

              We thank the Reviewer for knowledgeable comments and guidance to improve our manuscript.  We provide below point-by-point response to each comment.  Changes made to the text and new references are in blue.

Comments and Suggestions

This case control study focused on diabetes patients and evaluated effects of cadmium and diabetes on kidney outcomes. Although with a modest population, the authors performed comprehensive analysis and concluded novel findings that are clinically significant. In addition to the advantages, there are few questions need to answer:

1). The sample size of diabetic patients is limited, especially consider the number of patients with reduced eGFR. Are the patients with reduced eGFR undertaking specific treatments that could affect the results?

RESPONSE: We construct a new paragraph below to clarify that results are interpretable although the number of diabetes subjects is modest (n = 88), and have insert it in the Discussion (lines 254- 262) together with three additional references.

“The percentage of a reduced eGFR in controls, <10-y DM and ≥ 10-y DM groups were 11.4%, 14.6% and 27.0%, respectively (Table 1). The prevalence of a reduced eGFR in the controls of 11.4% is higher than that reported in studies from Spain (7%) [28] and Taiwan (6.3%) [29], but in line with the global prevalence of CKD, which varies between 8% and 16% [30]. None of diabetic cases with reduced eGFR received special treatments other than antidiabetic medication (metformin) [31]. Due to a uniform treatment and uniform Cd exposure levels evident from the similar blood and urinary excretion rate across three groups, the higher number of cases with a reduced eGFR in the >10-y DM, compared with the < 10-y DM groups could be attributable to Cd and diabetes.”

[28] Grau-Perez, M.; Pichler, G.; Galan-Chilet, I.; Briongos-Figuero, L.S.; Rentero-Garrido, P.; Lopez-Izquierdo, R.; Navas-Acien, A.; Weaver, V.; García-Barrera, T.; Gomez-Ariza, J.L.; et al. Urine cadmium levels and albuminuria in a general population from Spain: A gene-environment interaction analysis. Environ. Int. 2017, 106, 27-36.

[29] Tsai, H.J.; Hung, C.H.; Wang, C.W.; Tu, H.P.; Li, C.H.; Tsai, C.C.; Lin, W.Y.; Chen, S.C.; Kuo, C.H. Associations among heavy metals and proteinuria and chronic kidney disease. Diagnostics 2021, 11, 282.

[30] Kalantar-Zadeh, K.; Jafar, T.H.; Nitsch, D.; Neuen, B.L.; Perkovic, V. Chronic kidney disease. Lancet 2021, 398, 786-802.

[31] Sonthon, P.; Promthet, S.; Changsirikulchai, S.; Rangsin, R.; Thinkhamrop, B.; Rattanamongkolgul, S.; Hurst, C.P. The impact of the quality of care and other factors on progression of chronic kidney disease in Thai patients with type 2 diabetes mellitus: A nationwide cohort study. PLoS ONE 2017, 12, e0180977.

2). What type of therapy treatment is given to the diabetes patients in the study? What could be the result if there are differences during treatment? If patients undertake dialysis, will there be any changes regarding the association between Cd and kidney function?

RESPONSE: Diabetic cases were uniformly treated with antidiabetic medication. This is a common practice in Thailand based on a nationwide cohort study showing that glycemic control is the most effective treatment to protect against progression of CKD among diabetes patients [31]. Thus, it is likely that kidney outcome observed were due to Cd and/or diabetes, as stated in response 1 above.

3).  Please provide reference indicating the cutoff number of the level of (Eβ2M/Ccr) ×100 regarding tubular dysfunction and severe tubular dysfunction in Table 3 and Table 4. The number of patients defined with severe tubular dysfunction is not summarized in the characteristic table.

RESPONSES:

  • The references for the cutoff values for urinary excretion of β2M that have been used in health risk assessment of Cd in the human diet have been provided (lines 108-110 and lines 116-118).
  • Data on the severity of tubular dysfunction have been added to Table 1.

[25] Satarug, S.; Vesey, D.A.; Gobe, G.C. Dose–Response Analysis of the Tubular and Glomerular Effects of Chronic Exposure to Environmental Cadmium. Int. J. Environ. Res. Public Health 2022, 19, 10572.

[26] JECFA. Evaluation of certain Food Additives and Contaminants. In Proceedings of the Seventy-third meeting of the Joint FAO/WHO Expert Committee on Food Additives, Geneva, Switzerland, 8–17 June 2010; Food and Agriculture Organization of the United Nations; World Health Organization: Geneva, Switzerland, 2010. Available online: https://apps.who.int/iris/handle/10665/44521 (accessed on 17 May 2023).

[27] Wong, C.; Roberts, S.M.; Saab, I.N. Review of regulatory reference values and background levels for heavy metals in the human diet. Regul. Toxicol. Pharmacol. 2022, 130, 105122.

4).  Table 3 indicates the POR of microalbuminuria is associated with DM duration whereas in table 5 both hypertension and eGFR levels are shown to be associated with microalbuminuria. How to interpret the differences?

RESPONSES:

  • To clarify the results of the logistic regression for microalbuminuria, we have changed the title of table 5 as below.

Table 5. Effects of hypertension and GFR on prevalence odds of microalbuminuria.

  • The logistic regression of microalbuminuria in Table 5 incorporated three eGFR levels and hypertension as independent variables, where the distribution of diabetics with hypertension were compared across three eGFR levels. The results indicate that risk of microalbuminuria was affected by hypertension and eGFR levels.
  • We examined these results further with univariate/covariance analysis (Figure 3), where a significant increase in albumin excretion was seen only in the diabetics who had also hypertension.
  • Because of this effect of hypertension in a specific subgroup, a logistic regression of microalbuminuria in Table 3 that incorporated the duration of diabetes failed to detect it.

5). Figure 2 indicates Log [(Eβ2M/Ccr) ×103] is inversely associated with eGFR in both control and diabetics patients whereas Log [(EAlb/Ccr) ×104] dose not. What could be the reason of the differences? Does it suggest that Eβ2M/Ccr is a better indicator of eGFR level?

RESPONSE:

  • We agree with the reviewer that our finding argue strongly that Eβ2M/Ccr is a better indicator of eGFR level.
  • We provided additional explanations in the Discussion on albumin excretion rate to illustrate differences between β2M and Alb (lines 295-304).

“Albumin is not normally filtered by glomeruli, due to its large molecular weight and its negative charge. The small amount of albumin which does enter the urinary space is reabsorbed almost completely and a fraction of this is returned to the systemic circulation [35,36]. In an experimental study, Cd was found to disable the cubilin/megalin receptor system of albumin endocytosis, leading to albuminuria [37]. In addition, Cd diminished expression of megalin and ClC5 channels [38]. Cd may also increase glomerular permeability to albumin, as shown in another study, where a non-cytotoxic concentration of Cd (1 µM) increased the permeability of human renal glomerular endothelial cells in monolayers and caused the redistribution of the adherens junction proteins, vascular endothelial-cadherin and β-catenin [39,40].

[35] Molitoris, B.A.; Sandoval, R.M.; Yadav, S.P.S.; Wagner, M.C. Albumin uptake and processing by the proximal tubule: Physiological, pathological, and therapeutic implications. Physiol. Rev. 2022, 102, 1625–1667.

[36] Gburek, J.; Konopska, B.; Gołąb, K. Renal handling of albumin-from early findings to current concepts. Int. J. Mol. Sci. 2021, 22, 5809.

[37] Santoyo-Sánchez, M.P.; Pedraza-Chaverri, J.; Molina-Jijón, E.; Arreola-Mendoza, L.; Rodríguez-Muñoz, R.; Barbier, O.C. Impaired endocytosis in proximal tubule from subchronic exposure to cadmium involves angiotensin II type 1 and cubilin receptors. BMC Nephrol. 2013, 14, 211.

[38] Gena, P.; Calamita, G.; Guggino, W.B. Cadmium impairs albumin reabsorption by down-regulating megalin and ClC5 channels in renal proximal tubule cells. Environ. Health Perspect. 2010, 118, 1551-1556.

[39] Li, L.; Dong, F.; Xu, D.; Du, L.; Yan, S.; Hu, H.; Lobe, C.G.; Yi, F.; Kapron, C.M.; Liu, J. Short-term, low-dose cadmium exposure induces hyperpermeability in human renal glomerular endothelial cells. J. Appl. Toxicol. 2016, 36, 257-265.

[40] Li, Z.; Jiang, L.; Tao, T.; Su, W.; Guo, Y.; Yu, H.; Qin, J. Assessment of cadmium-induced nephrotoxicity using a kidney-on-a-chip device. Toxicol. Res. 2017, 6, 372-380.

Comments on the Quality of English Language

There are some minor spelling errors ("Alb excretion rats", page 12) and repeated content ("These findings" in abstract)

RESPONSE:  The referred typo errors have been corrected. 

Round 2

Reviewer 1 Report

No further comments.